# New Method for Sugarcane (*Saccharum* spp.) Variety Resources Evaluation by Projection Pursuit Clustering Model

**Yong Zhao** **, Yuebin Zhang, Jun Zhao, Fenggang Zan, Peifang Zhao, Jun Deng, Caiwen Wu and Jiayong Liu ***

Sugarcane Research Institute, Yunnan Academy of Agricultural Sciences, Kaiyuan 661699, China; 18087395132@163.com (Y.Z.); ynzyb@sohu.com (Y.Z.); junzhao_ky@126.com (J.Z.); fengang88@126.com (F.Z.); hnzpf@163.com (P.Z.); dj@yaas.org.cn (J.D.); gksky_wcw@163.com (C.W.)
* Correspondence: lljjyy1976@163.com

**Abstract:** In the breeding of new sugarcane varieties, the survey data do not always conform with a normal or linear distribution. To apply non-normal or non-linear data to evaluate new material requires a suitable evaluation model or method. The projection pursuit clustering (PPC) model is a statistical method that does not require making normal assumptions or other model assumptions on sample data, and is suitable to analyze high-dimensional, non-linear, and non-normal data. However, this model has been applied infrequently to crop variety evaluation. In this study, 103 varieties that have been bred over the last 70 years in China were planted, and their main industrial and agronomic traits were collected. Through the exploratory analysis of the data structure characteristics, the PPC model was used to evaluate these sugarcane varieties. The model provided good projection directions of agronomic and industrial traits, with accurate projection values. PPC models could evaluate sugarcane resources well, and the results were objective and reliable. Thus, the PPC model could be used as a new method for crop variety evaluation. At the same time, 51 excellent industrial and agronomic variety resources were screened for application in breeding.

**Keywords:** sugarcane variety; projection pursuit clustering; PPC; agronomic/industrial trait; projection direction; projection value



## 1. Introduction

Statistical analysis of high-dimensional data is becoming increasingly common and important, in which multivariate analysis is a powerful tool to solve the problems of high-dimensional data. The traditional multivariate analysis method assumes that the population obeys a normal distribution; however, many data in practical problems do not meet the assumption of normality, and need to be solved using robust or nonparametric methods [1]. To solve this problem, the international statistical community proposed an exploratory data analysis method that begins by examining the data directly, then analyzing the simulated data using a computer, and finally designing a software program test. Projection pursuit (PP) is an effective method to realize this new thinking [2], which was developed by the international statistical community in the mid-1970s. It is an interdisciplinary fusion of statistics, applied mathematics, and computer technology. This statistical method is used to analyze and process high-dimensional observation data, especially non-normal and non-linear data. By projecting the high-dimensional data onto the low-dimensional subspace, it finds the projection that can reflect the structure or characteristics of the original high-dimensional data, to permit the study and analysis of high-dimensional data [3]. The basic idea is to use computer technology to project the high-dimensional data into the low-dimensional (1–3-dimensional) subspace through some combination, find the projection that can reflect the original high-dimensional data structure or characteristics by minimizing a projection index, and analyze the data structure in the low-dimensional space, thus allowing high-dimensional data study and analysis [4].

Its general scheme comprises first selecting a distribution model as the standard (generally a normal distribution), and considering it the least interesting structure. Secondly, the data is projected into the low dimensional space to find the projection with the largest difference between the data and the standard model, which shows that the projection contains the structure that the standard model cannot reflect. Third, the structure contained in the above projection is removed from the original data to obtain improved new data. Finally, the second and third steps are repeated for the new data until there is no significant difference between the data and the standard model in any projection space. The PP method can reduce the problem of dimensionality to a great extent, because its data analysis is carried out in a low-dimensional subspace [5]. For a 1–3-dimensional projection space, sparse data points in the high-dimensional space are sufficient to find the structural characteristics of the data in the projection space. It has the advantages of robustness, anti-interference, and high accuracy, and has thus been used widely in many fields [6].

Projection pursuit clustering (PPC) is a clustering and classification analysis method that can be used for both exploratory analysis and deterministic analysis [7]. The so-called projection is essentially to observe the data from different angles and find the optimal projection direction that can reflect the data characteristics to the greatest extent, and then fully mine the data. PPC is an effective dimensionality reduction technology that can be used for high-dimensional data analysis, especially suited for non-linear and non-normal problems [8]. The evaluation results have a high coincidence rate with the actual situation. It has been used widely in water quality evaluation [9], the comprehensive evaluation of atmospheric environmental quality [10], disaster assessment [11], and enterprise competitiveness [12]. PPC is characterized by projecting the high-dimensional data onto the low-dimensional (1–3-dimensional) subspace when the weight coefficient is unknown [13]. For the projected configuration, the projection index function is used to measure the possibility of the projection exposing a certain structure, and to find the projection value that makes the projection index function reach the optimal. Then, the structural characteristics of the high-dimensional data are analyzed according to the projection value, or a mathematical model is constructed according to the scatter diagram between the projection value and the output value of the research system to predict the output of the system [7]. It avoids the human interference factors of expert scoring, saves the steps of using expert scoring and evaluation, and is more accurate and convenient. Therefore, it has advantages in the processing of quantitative evaluation index data.

Sugarcane (*Saccharum* spp.) is a gramineous $C_4$ plant, and an important raw material of global sugar production. Sugarcane cross-breeding in China began with the establishment of the Hainan sugarcane breeding farm in 1953 [14]. In the last 70 years, more than 100 sugarcane varieties have been selected, approved, and named in China [15]. In the process of sugarcane variety improvement, many sugarcane varieties have been replaced in sugarcane planting regions, and the main varieties have been continuously updated. Some of them have dominated the planting area for decades, some have died out in the process of popularization, and some have been seriously degraded and were phased out in the later stage. The important function of sugarcane variety improvement research is sugarcane cross-breeding. The improvement of sugarcane varieties in China has experienced five generations [16]. The first generation is represented by bamboo cane and reed cane; the second generation is represented by F134; the third generation is represented by GT11 and YZ71388; the fourth generation is represented by ROC22, YT93159, GT35, and others; and the fifth generation is represented by LC05136, YZ081609, YZ 0551, GT42, and others.

At present, the core function of sugarcane cross-breeding is still to improve and enhance sugarcane agronomic and industrial traits, especially the yield characteristics and sucrose quality traits. The replacement of sugarcane varieties is usually carried out using breeding or the introduction of new materials that are better than the original varieties. The success of sugarcane cross-breeding is based on the scientific selection of cross combinations. The combination of sugarcane hybrids is inseparable from the identification and evaluation of their parents or germplasm. Therefore, the collection, identification, evaluation, and construction of sugarcane parents is an important guarantee to improve breeding efficiency. At present, the research on germplasm evaluation mainly focuses on field phenotype investigation and genetic diversity analysis [17], the development and application of molecular markers [18], and physiological and biochemical analyses [19]. The investigation and analysis of field phenotype data is mainly based on comprehensive analysis methods, such as principal component analysis [20], whereas the PPC method is less used in crop evaluation and analysis. We conducted this experiment mainly based on the PPC model to comprehensively evaluate and analyze cultivated sugarcane varieties since the development of sugarcane cross-breeding in China. We hypothesized that the application of PPC would introduce new evaluation methods and provide support for the accurate evaluation of sugarcane germplasm. At the same time, we screened excellent variety resources for reference and application in breeding.

## 2. Materials and Methods

### 2.1. Experimental Materials

A total of 103 sugarcane varieties were tested, which have been bred from 1953 to 2010 (Table 1). Many of them were the main cultivated varieties in different sugarcane development periods. Among them, there are 26 sugarcane varieties of "Yunzhe" series, 2 varieties of the "Chuantang" series, 3 varieties of the "Dezhe" series, 4 varieties of the "Funong" series, 11 varieties of the "Ganzhe" series, 2 varieties of the "Gannan" series, 2 varieties of the "Hainan" series, 4 varieties of the "Liucheng" series, 3 varieties of the "Mintang" series, 10 varieties of the "Yuetang" series, and 1 variety of the "Yuegan" series; plus 16 varieties of the "Guitang" series, 15 varieties of the "ROC" series, and 4 varieties of the "F" series that were imported from Taiwan.

**Table 1.** Information of test varieties series and the breeding institution.

| Variety Series [1] | Breeding Institution | Material Quantity |
|---|---|---|
| YZ | Sugarcane Research Institute of Yunnan Academy of Agricultural Sciences | 26 |
| CT | Plant Engineering Research Institute of Sichuan Province | 2 |
| DZ | Sugarcane Research Institute of Dehong Prefecture, Yunnan Province | 3 |
| FN | Fujian Agriculture and Forestry University | 4 |
| GZ, GN | Gannan Academy of Sciences | 13 |
| HN | Former South China Institute of Agricultural Sciences | 2 |
| LC | Guangxi Liucheng Institute of Agricultural Sciences | 4 |
| MT | Fujian Academy of Agricultural Sciences | 3 |
| YT, YG | Biological Engineering Institute of Guangdong Academy of Sciences | 11 |
| ROC | Materials imported from Taiwan, China | 15 |
| F | Materials imported from Taiwan, China | 4 |
| GT | Sugarcane Research Institute of Guangxi Academy of Agricultural Sciences | 16 |

[1] Abbreviations of sugarcane varieties: "Yunzhe" is referred to as "YZ", "Chuantang" is referred to as "CT", "Dezhe" is referred to as "DZ", "Funong" is referred to as "FN", "Ganzhe" is referred to as "GZ", "Gannan" is referred to as "GN", "Huanan" is referred to as "HN", "Liucheng" is referred to as "LC", "Mintang" is referred to as "MT", "Yuetang" is referred to as "YT", "Yuegan" is referred to as "YG", Xintaitang is referred to as "ROC", "Guitang" is referred to as "GT".

## 2.2. Experimental Site and the Test Design

A field experiment was arranged in the main scientific research base of the Sugarcane Research Institute, Yunnan Academy of Agricultural Sciences (Kaiyuan City, Yunnan Province; 23.71° N, 103.25° E) on 20 December 2016. The test soil type was clay, organic matter content, 19.7 g/kg; pH value 7.9; available phosphorus 55.1 mg/kg; available potassium 56.0 mg/kg; alkali hydrolyzed nitrogen, 77.1 mg/kg. A randomized complete block design was applied in our experiment with two replications. 103 varieties were evaluated, and each variety was planted with a length of 8.0 m, and a row spacing of 1.1 m. Each variety was planted in 10 rows with protective rows that occupied 88 m$^2$ plot area of each replication. In 2016, we used sugarcane buds to generate the new plant, and this stage was called New Plant (December 2016–March 2018). After sugarcane stems developed from the remaining roots of 1st year were harvested, we called the stage Ratoon 1 (April 2018–March 2019). After harvesting the sugarcane of Ratoon 1, the left root in the soil grew up and the stems were gathered, which we called Ratoon 2 (April 2019–March 2020). In this experiment, we collected the data of characteristics for 3 years, including new plant, ratoon 1, and ratoon 2. The experimental plot was irrigated, and the management was consistent with standard field production practices. The agronomic and industrial traits of sugarcane were collected at maturity in December of 2017 (new plant), 2018 (ratoon 1), and 2019 (ratoon 2).

### 2.2.1. Agronomic Characters

The investigation of agronomic traits adopted methods of graded data referring to previously published research [21], and the range of agronomic traits was set mainly based on the grading requirements of this experiment, combined with breeding experience. The survey was divided into three periods: new planting, ratoon 1, and ratoon 2, and the survey time was the beginning of December each year. First, a grading team composed of 3–4 sugarcane breeders (during the grading industrial traits, the members of the grading team remained unchanged until the grading work was completed) selected five important agronomic character traits, including plant height, stem diameter, millable stalks, leaf and main field diseases (comprehensively reflecting natural field diseases such as mosaic disease, brown rust, smut, and pokkah boeng), and general vigor. Second, each trait was divided into five grades. For plant height, the highest was grade 1, and grade 5 was the shortest. For stem diameter, the thickest was grade 1, and grade 5 was the thinnest. The most millable stalks were grade 1, and grade 5 were the least millable. For leaf diseases, grade 1 was none or light, whereas grade 5 was the most serious disease. For general vigor, grade 1 grew best, and grade 5 showed the worst growth. During grading, the grading team referred to the numerical range of each character, as shown in Table 2. The classification of leaf disease and field main diseases was determined according to the disease performance of mosaic disease, brown rust, smut, and pokkah boeng. The weight of a single sugarcane stem was measured directly. Each sugarcane variety was randomly sampled as six sugarcane stems to form the test sample (three sugarcane stems for each replication), and then, average value was calculated.

**Table 2.** The reference grading range of agronomic traits.

| Rank | Sugarcane Height/cm | Stem Diameter/cm | Millable Stalks Number/m$^2$ | Leaf Disease/% | General Vigor |
|---|---|---|---|---|---|
| 1 | >280 | >3.00 | >10 | <5 | According to the observation, the |
| 2 | 220–280 | 2.5–3.00 | 8–10 | 5–10 | overall performance of |
| 3 | 160–220 | 2.00–2.50 | 6–8 | 10–20 | germplasm resources was |
| 4 | 100–160 | 1.50–2.00 | 4–5 | 20–30 | comprehensively evaluated and |
| 5 | <100 | <1.50 | <4 | >30 | graded |

The grade range refers to the evaluation standard of sugarcane agronomic characters in China, and the long-term selection experience of breeders.

### 2.2.2. Industrial Traits

As industrial characteristics, sugarcane brix, the sucrose content, juice sucrose content, fiber content, and gravity purity were determined with reference to a previous report [22]. Juice was extracted from three stalks selected randomly from every plot using a mechanical cane juicer in all environments. The extracted juice was detected for brix (%) using an automatic refractometer, Rudolph J257 (Rudolph Research Analytical, Hackettstown, NJ, USA), and the sucrose content (%), juice sucrose content (%), and gravity purity (%) was measured using an automatic saccharimeter, Autopol 880 (Rudolph Research Analytical, Hackettstown, NJ, USA). After extracting juice, the remaining cane residue was weighed and oven-dried to determine the fiber content (%). New plantings were sampled once during the middle ten days of a month from November 2017 to March 2018, ratoon 1 was sampled monthly from November 2018 to March 2019, and ratoon 2 was tested monthly from November 2019 to March 2020.

### 2.2.3. Climate Conditions of Test Site

The Yunnan climate belongs to the subtropical plateau monsoon type, with remarkable three-dimensional climate characteristics, distinct dry and wet seasons, and abnormal changes in temperature with vertical terrain. The experimental site (23.7° N, 103.25° E) had an altitude of 1051.8 m, and belongs to the subtropical plateau monsoon climate, with sufficient light resources. See Table 3 for the detailed climatic parameters.

**Table 3.** Climate data in different years.

| Years | Sunshine/h | Average-Temperature/°C | Maximum Temperature/°C | Minimum Temperature/°C | Average-Rainfall/mm | Potential Evaporative/mm | Frost-Free Period/d |
|-------|-----------|------------------------|------------------------|------------------------|---------------------|--------------------------|---------------------|
| 2017 | 1960 | 20.1 | 34.1 | 3.3 | 1038.4 | 1987 | 341 |
| 2018 | 2125.3 | 21.5 | 37.7 | 0.2 | 698.2 | 1880 | 320 |
| 2019 | 2033.7 | 20.8 | 36.2 | 4.2 | 592 | 1860 | 330 |

### 2.3. Method and Principle of Projection Pursuit Clustering (PPC)

The calculation steps of the PPC method are as follows:

(1) Normalization of the sample evaluation index set used the calculation formulas:

$$x(i,j) = \frac{x^*(i,j) - x_{min}(j)}{x_{max}(j) - x_{min}(j)}$$

$x^*(i,j)$, $i = 1, \ldots, n$; $j = 1, \ldots, p$; where the "$j$" is the evaluation index value of the "$i$" sample; "$n$" and "$p$" are the number of samples and the number of evaluation indicators, respectively; and $x_{max}(j)$ and $x_{min}(j)$ are the maximum and minimum values of the "$j$" evaluation index in the sample set.

(2) Construction of the projection objective function Q($a$). The essence is to synthesize the p-dimensional data $x$ ($i, j$), $j = 1, \ldots, p$ into a one-dimensional projection value $z(i)$ with $a = (a(1), a(2), \ldots, a(p))$ as the projection direction.

$$z(i) = \sum_{i=1}^{n}(a(j)x(i,j), i = 1, \ldots, P)$$

where "$a$" is the unit length vector.

When synthesizing the projection values, it is required that the dispersion characteristics of the projection value $z(i)$ are: the local projection points should be as dense as possible, preferably condensed into several point clusters; and the projection point clusters should be as scattered as possible as a whole. Based on this, the projection index function can be constructed as:

$$Q(a) = S_Z D_Z$$

where $S_Z$ is the standard deviation of the projection value $z(i)$, and $D_Z$ is the local density of the projection value $z(i)$. The calculation formulae are:

$$S_Z = \sqrt{\sum_{i=1}^{n} z(i) - \bar{z}/(n-1)}$$

$$D(z) = \sum_{i=1}^{n} \sum_{j=1}^{n} (R - r_{ij}) u(R - r_{ij})$$

$$r_{ij} = |z(i) - Z(j)|$$

where "$a$" is the unit length vector; "$z$" is the mean of series $z(i)$, $i = 1, \ldots, n$; $R$ is the window radius for local density. "$u_t$" is the unit step function. When t $= (R - r_{ij}) \geq 0$, the function value is 1, and when t < 0, the function value is 0.

(3) The optimal projection direction is estimated by maximizing the projection index function to reveal a certain kind of feature structure of the high-dimensional data. The calculation formula is:

$$maxQ(\prime a) = S_z D_z$$

$$s.t \sum_{l=1}^{p} a^2(j) = 1, a(j) \geq 0$$

where "$s.t$" stands for the constraint condition, which is a complex nonlinear optimization problem with $a(j)$, $j = 1, \ldots, p$ as optimization variables. It is generally solved by a genetic algorithm simulating the survival of the fittest rules and the chromosome information exchange mechanism within the population.

(4) Using the formula $maxQ(\prime a) = S_z D_z$ and $s.t \sum_{l=1}^{p} a^2(j) = 1, a(j) \geq 0$, the optimal projection direction "$a$*" is substituted into the formula $z(i) = \sum_{i=1}^{n} (a(j)x(i,j))$, thus the projection value "$z(i)$" of each evaluation sample is obtained, which are sorted from large to small, allowing the evaluation index sample set to be evaluated uniformly.

### 2.4. Data Analysis

The grading data of agronomic traits (countdown processing) and the metric data of industrial traits were adopted in the PPC model. Microsoft Excel 2019 (Microsoft, Redmond, WA, USA) was used for data collecting and sorting; DPS v14.10 (data analysis software, Zhejiang University, Hangzhou, China) was used for variance analysis, and calculating the projection direction and projection value. R software [23] was used for data processing and figure creation. We used the "ggplot2" to draw distribution characters of the projection values, column and scatter diagram chart, and box diagram; "ggplot2", "ggpubr", and "ggpmisc" were used for analysis of correlation; "ggtree" was used for cluster analysis.

## 3. Results
### 3.1. Frequency Distribution Analysis of the Agronomic and Industrial Traits

The grading data of agronomic traits were discontinuous data and did not accord with the characteristics of a normal distribution. Sugarcane sucrose, sugarcane brix, sugarcane fiber, and gravity purity basically conformed to the characteristics of a normal distribution, except juice sugar contents (Figure 1).

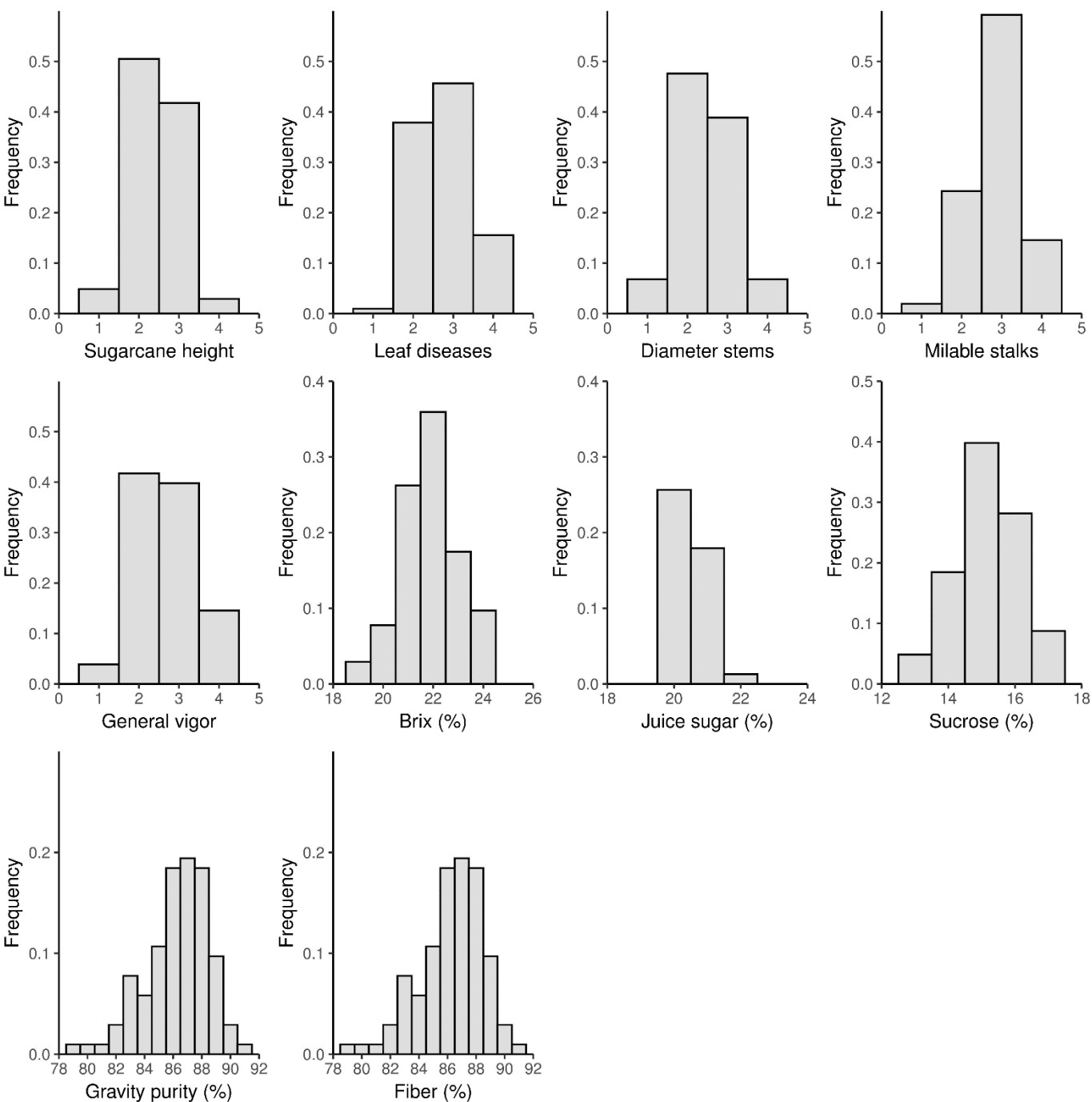

**Figure 1.** Frequency distribution analysis of the 10 agronomic and industrial traits data.

### 3.2. Analysis of the Correlation of Certain Traits

Sugarcane brix, juice sucrose, and cane sucrose correlated significantly ($p < 0.01$) (Figure 2). Therefore, in the process of sugarcane variety breeding, sugarcane brix can predict the content of sucrose of a variety.

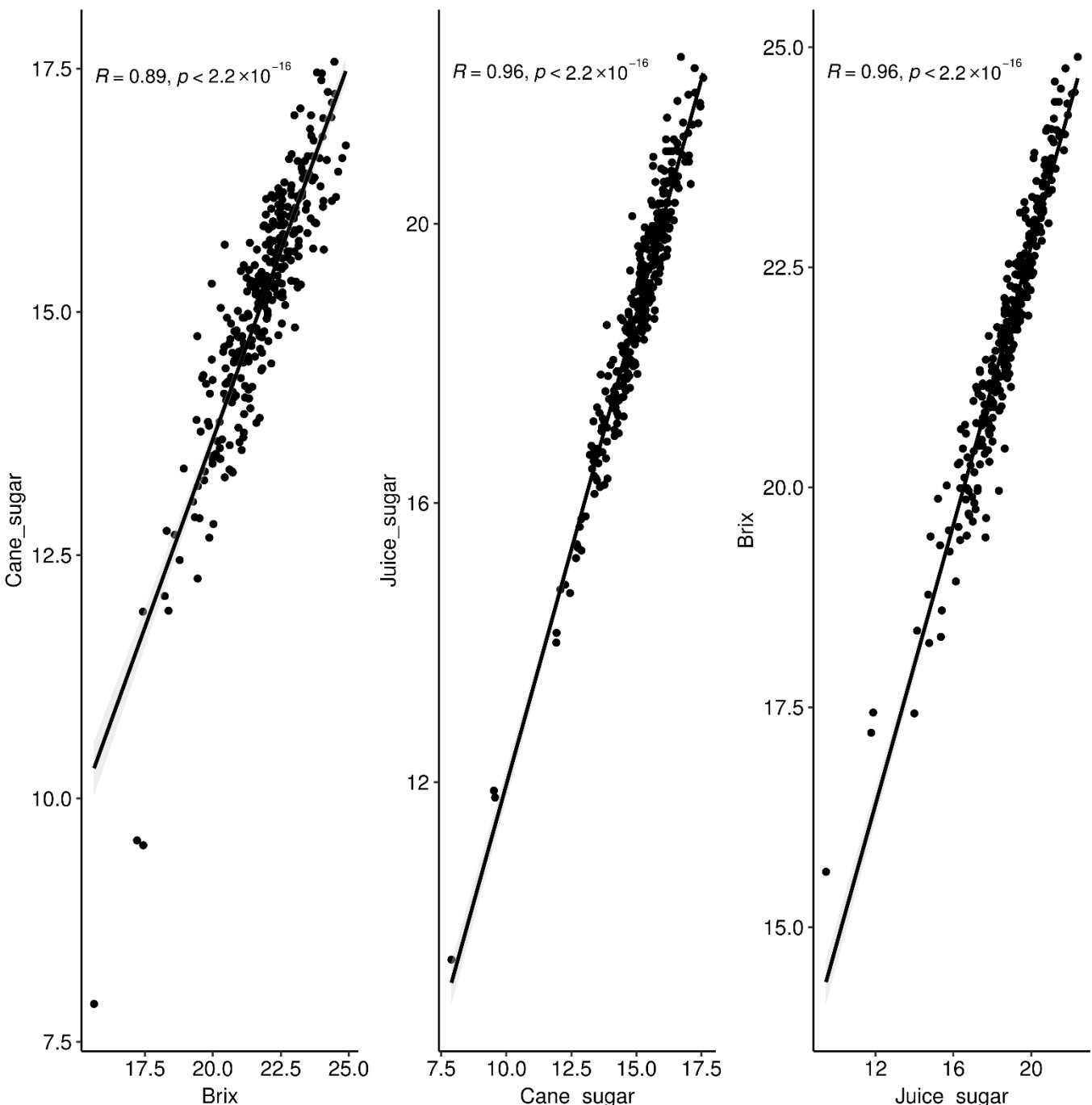

**Figure 2.** Analysis of the correlation of certain traits by Pearson correlation analysis.

*3.3. Variance Analysis of the Agronomic and Industrial Traits*

Variance analysis of the 103 sugarcane varieties showed that there were significant differences for ten traits in the new planting and ratoon periods ($p < 0.05$), except for sugarcane height in ratoon 2. The coefficients of variation among the agronomic traits were 27.15–41.52%, with wide variations, and the coefficients of variation among the industrial traits were 3.27–8.30% (Table 4).

**Table 4.** Variance analysis on agronomic and industrial traits of the 103 sugarcane varieties.

| Sugarcane Traits | Plant | | Ratoon 1 | | Ratoon 2 | | Mean | |
|---|---|---|---|---|---|---|---|---|
| | Average Values | Variation Coefficient/% | Average Values | Variation Coefficient/% | Average Values | Variation Coefficient/% | Average Values | Variation Coefficient/% |
| Height | 2.32 ** | 34.72 | 2.72 ** | 36.56 | 2.33 | 34.14 | 2.45 ** | 36.20 |
| Stem diameter | 2.50 ** | 34.22 | 2.60 ** | 36.74 | 2.42 ** | 33.16 | 2.51 ** | 34.91 |
| Millable stalks | 2.96 ** | 28.01 | 2.99 ** | 33.68 | 2.83 ** | 27.15 | 2.93 ** | 29.91 |
| Leaf disease | 2.85 ** | 34.69 | 2.98 ** | 41.52 | 2.70 * | 32.66 | 2.84 ** | 36.96 |
| General vigor | 2.73 ** | 35.36 | 2.92 ** | 37.74 | 2.52 * | 34.93 | 2.72 ** | 35.04 |
| Brix (%) | 21.35 ** | 6.47 | 21.78 ** | 5.63 | 22.54 ** | 5.51 | 21.89 ** | 6.27 |
| Juice sugar (%) | 18.19 ** | 8.92 | 18.86 ** | 6.92 | 19.74 ** | 6.90 | 18.93 ** | 8.30 |
| Cane sugar (%) | 14.80 ** | 8.12 | 15.02 ** | 6.23 | 15.69 ** | 6.06 | 15.17 ** | 7.27 |
| Purity (%) | 84.84 ** | 3.86 | 86.40 ** | 2.49 | 87.47 ** | 2.59* | 86.24 ** | 3.27 |
| Fiber (%) | 13.45 ** | 13.69 | 15.18 ** | 12.56 | 15.38 ** | 12.98 | 14.67 ** | 14.31 |

**, $p < 0.01$; *, $p < 0.05$.

### 3.4. PPC Applied to the Comprehensive Evaluation of Sugarcane Varieties

According to the formula calculated in the DPS v14.0 statistical analysis software, the PPC model was used to analyze the data of 10 agronomic characters of 103 sugarcane varieties. From the frequency distribution verification analysis (Figure 1), we observed that the grading data of agronomic traits were discontinuous and did not conform to normal characteristics, whereas the industrial traits were continuous and basically conformed to normal distribution characteristics. Therefore, the 10 industrial and agricultural traits were divided into two parts, namely, the PPC analysis of the data of agronomic traits, and the PPC data analysis of the industrial traits (Figure 3). In the PPC model, we mainly paid attention to the determination of the projection direction (can be understood as the weight of each trait to the comprehensive evaluation), and then calculated the projection values according to the projection direction (can be understood as the comprehensive score of varieties obtained according to the projection direction). The projection values represent the comprehensive performance of the varieties, which, together with the projection direction, indicated increased performance of the comprehensive agronomic traits. However, if the projection values were small, the performance of the comprehensive agronomic traits was worse.

### 3.4.1. The Agronomic Traits Grading Data Analyzed in the 103 Sugarcane Variety Resources

The projection direction of the millable stalks was the largest, followed by the traits of general vigor, plant height, and the stem diameter, and that of leaf disease was the smallest. According to the PPC principles, the projection direction represents the evaluation weight, thus the millable stalks had the highest evaluation weight, and the leaf disease had the lowest (Figure 3A1). The projection values of the 103 varieties were calculated by the projection direction. The projection values were evenly distributed and mainly concentrated in the −2.5 to 2.5 area (Figure 3A2).

### 3.4.2. The Industrial Traits Data Analyzed in the 103 Sugarcane Variety Resources

The projection direction of gravity purity was the largest, followed by the traits of cane sugar, juice sugar, and brix, and that of sugarcane fiber was the smallest. Cane sugar and juice sugar had the same direction (Figure 3B1). The projection values of the 103 varieties were calculated by the projection direction. The projection values were evenly distributed and mainly concentrated in the −5 to 5 area (Figure 3B2).

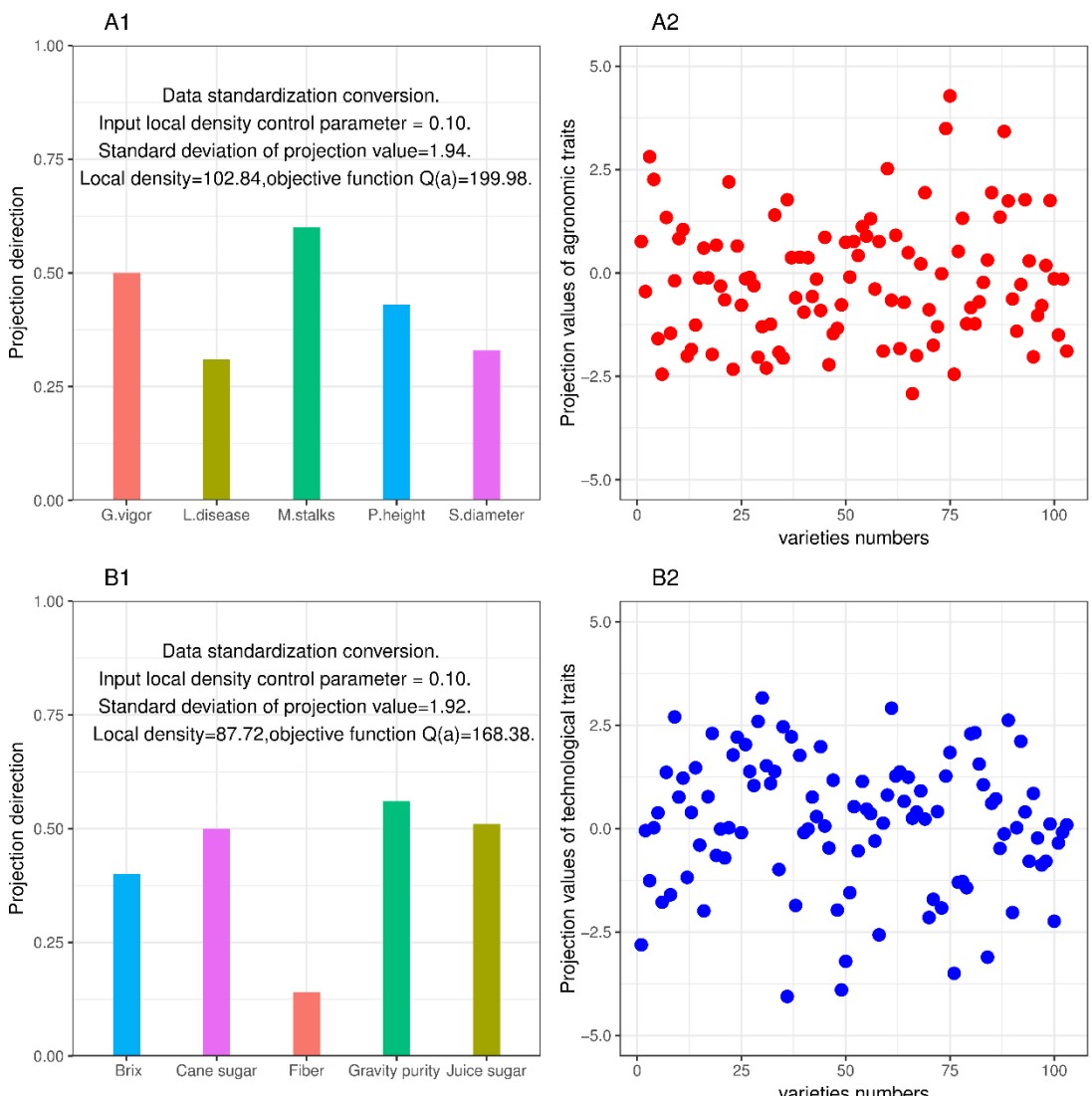

**Figure 3.** Agronomic and industrial traits of 103 sugarcane variety resources were evaluated and analyzed based on the projection pursuit clustering (PPC) model. (**A1**) is the visual column chart of agronomic projection direction, (**A2**) is the scatter diagram of agronomic projection values. (**B1**) is the visual column chart of industrial projection direction, (**B2**) is the scatter diagram of industrial projection values. G.vigor is the general vigor trait, L.disease is the leaf disease trait, M.stalks is millable stalks, P.height is the plant height trait, S.diameter is stem diameter.

### 3.4.3. Distribution Characters of the Projection Values among the 103 Sugarcane Varieties

The projection values of the comprehensive agronomic and industrial characters of the 103 sugarcane varieties were mainly distributed in four regions. These were the areas where the projection values of the industrial and agronomic traits were greater than 0, the areas where the projection values of the industrial and agronomic traits were less than 0, the areas where the projection value of the industrial traits was greater than 0 and the projection value of the agronomic traits was less than 0, and the areas where the projection value of the agronomic traits was greater than 0 and the projection value of the industrial traits was less than 0. This also showed that 103 sugarcane variety resources could be preliminarily divided into four categories: varieties with good comprehensive industrial and agricultural characters, varieties with poor industrial and agricultural characters, varieties with good industrial characters but poor agronomic characters, and varieties with good agronomic characters but poor industrial characters (Figure 4).

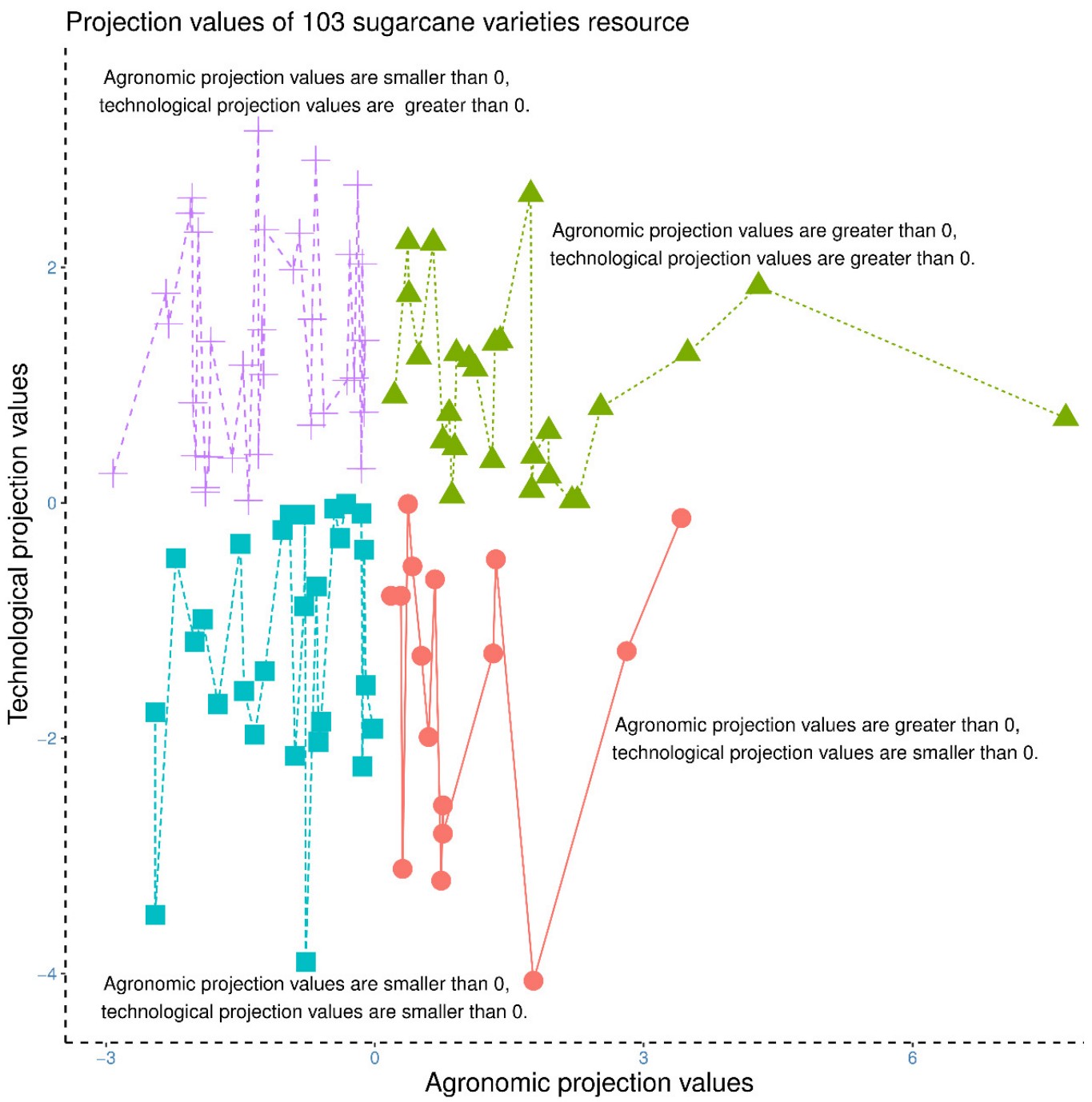

**Figure 4.** Projection values of 103 sugarcane variety resources.

### 3.5. Cluster Analysis of 103 Sugarcane Varieties Based on the Projection Values

To further evaluate and analyze the comprehensive qualities of the 103 sugarcane varieties, systematic cluster analysis was carried out according to the projection values of the industrial and agronomic characters (Figure 5). The 103 sugarcane varieties were divided into five groups, including three materials in group 1, 13 materials in group 2, 31 materials in group 3, 27 materials in group 4, and 29 materials in group 5. The projection values could well-separate the various varieties and gather them into these different groups.

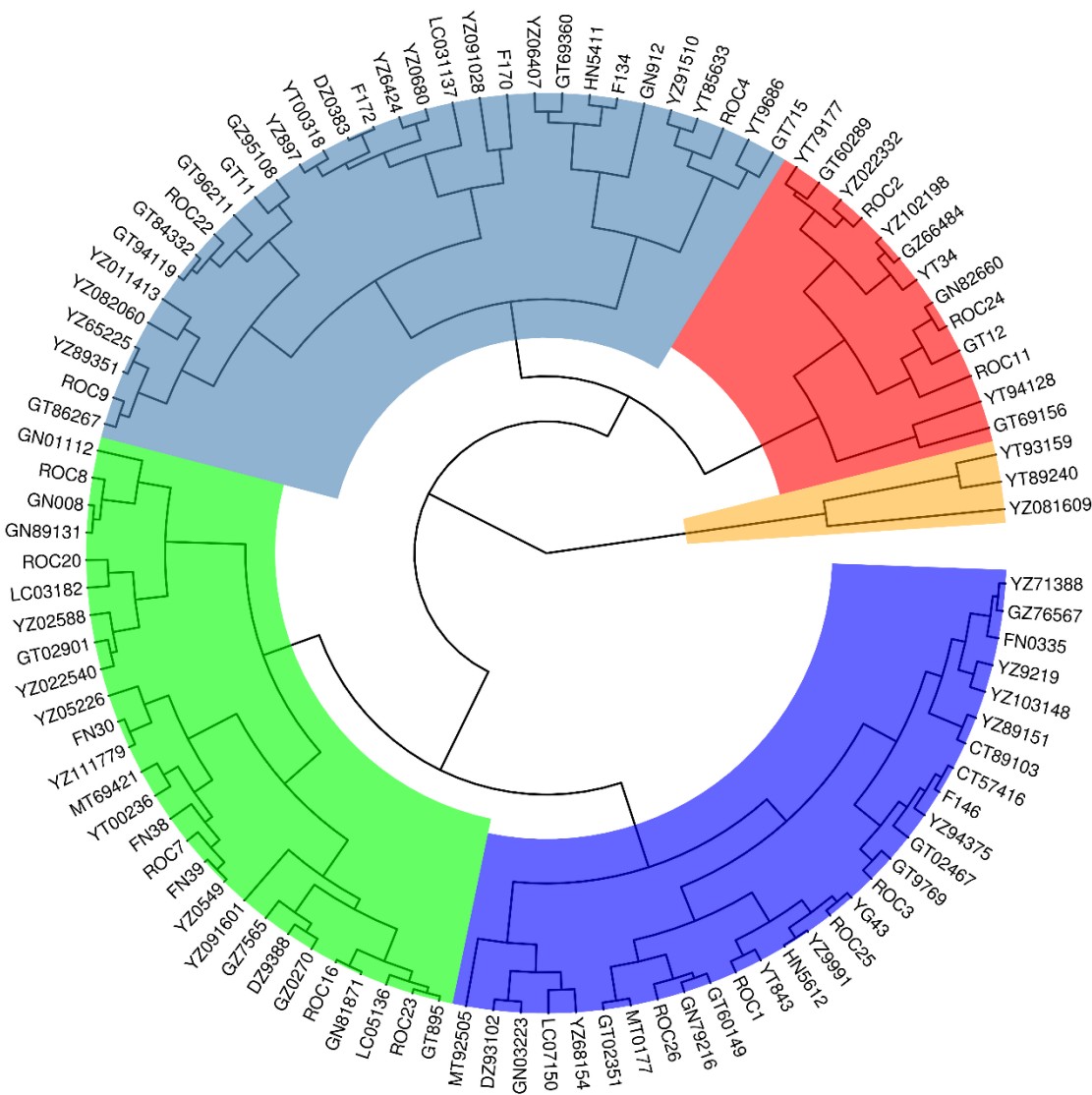

**Figure 5.** Cluster analysis of the projection values of 103 sugarcane variety resources. The different colors divide the different groups. The yellow region is group 1, the red region is the group 2, the grey region is group 3, the green region is group 4, and the blue region is group 5.

*3.6. Difference Analysis of Projection Values and Main Traits among the Cluster Groups*

The agronomic projection value of group 1 was the highest, followed by group 3 and group 4, and group 2 and group 5 were lower. The industrial projection values of group 1 and group 2 were higher, followed by group 5, group 2, and group 3 (Figure 6A). There was no significant difference in leaf disease between group 1, group 3, and group 4, and the values of these groups were low. This indicated that their leaf diseases occurrence rates were low. There was no significant difference between group 2 and group 5, but their values were higher (Figure 6B). There was no significant difference in general vigor between group 2 and group 5, whereas the other groups had significant differences. The value of group 1 was the lowest, followed by group 2 and group 4 (Figure 6C). This indicated that group 1 had the greatest growth potential, followed by group 2 and group 4, with group 3 and group 5 having the worst growth potential. There was no significant difference in sugarcane sugar between group 1, group 4, and group 5, and the three groups had significantly higher scores than group 2 and group 3 (Figure 6D). The results showed that the sugar content of sugarcane was the best in group 1, group 4, and group 5. There was no significant difference in gravity purity between group 1 and group 4; however, in these two groups, gravity purity was significantly higher than that in the other groups (Figure 6E).

Thus, group 1 and group 4 performed best for sugarcane gravity purity, whereas other groups showed poor performance. There was no significant difference in the sugarcane fiber content among the five groups (Figure 6F).

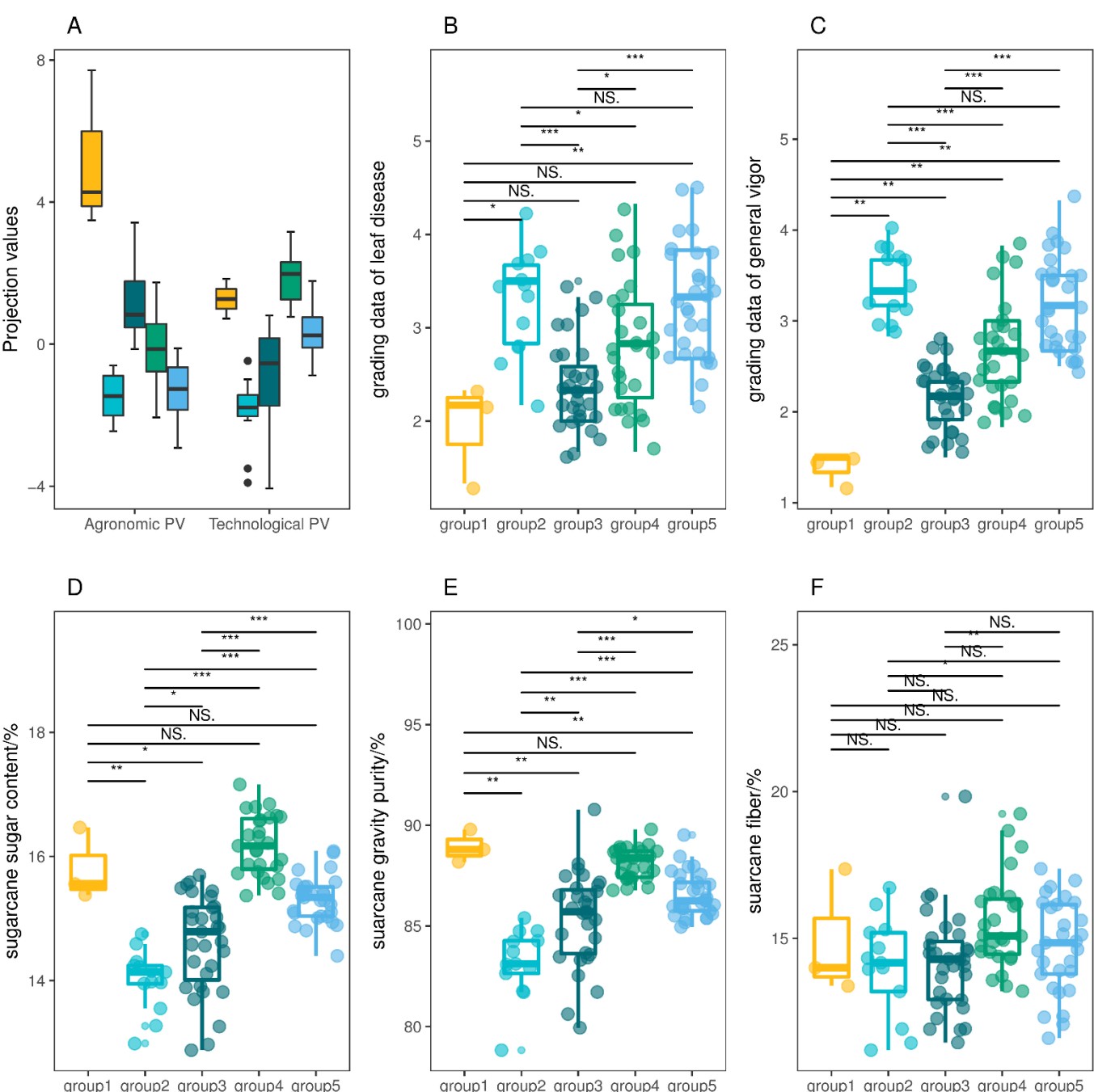

**Figure 6.** Difference analysis of the projection values. "PV" means projection value. Not significant (NS), $p > 0.05$; *, $p < 0.05$; **, $p < 0.01$; ***, $p < 0.001$. (**A**) is the agronomic and industrial projection values of 5 groups. (**B**) is projection values of leaf disease in different groups, (**C**) is projection values of general vigor in different groups. (**D**) is projection values of sugarcane sugar contents in different groups, (**E**) is projection values of sugarcane gravity in different groups, (**F**) is projection values of sugarcane fiber in different groups.

*3.7. Screening the Excellent Sugarcane Varieties Based on the PPC Models*

From the comparison of the different groups, it was found that group 1, group 3, and group 4 performed better in terms of sugarcane leaf diseases and sugarcane agronomic traits, and group 1, group 4, and group 5 performed better in terms of sugarcane industrial traits. Combined with the projection value of agronomic characters, we summarized and analyzed these materials, and screened sugarcane variety resources with better characters for reference. There were 26 variety resources for both the industrial and agricultural traits, which were mainly concentrated in group 1, group 3, and group 4. Their projection values of industrial and agricultural characters were all above 0, the main agronomic grading data were concentrated between 1 and 3, the sucrose content of sugarcane was more than 15%, and the gravity purity of sugarcane was more than 85%. There were four variety resources with excellent agronomic traits, but poor industrial traits. Their projection values for agronomic traits were more than 1, whereas their projection values for industrial traits were less than 0, and the cane sugar content was less than 15%. There were 20 sugarcane varieties with good industrial traits, but poor agronomic characters. Their projection values for industrial characters were more than 1, but their projection values for agronomic characters were less than 1, and their sugarcane sucrose content was about 16%; however, their performance for agronomic characters was poor (Table 5).

**Table 5.** Information of variety resources with excellent agronomic traits or industrial traits.

| Number | Varieties | Female Parent | Male Parent | Group | A_P_Value | T_P_Value | Height | Leaf Disease | Stem Diameter | Millable Stalks | General Vigor | Cane Sugar/% | Gravity Purity/% |
|---|---|---|---|---|---|---|---|---|---|---|---|---|---|
| 1 | YZ081609©® | YZ94343 | YT00236 | 1 | 7.71 | 0.72 | 1 | 1.33 | 1.17 | 1.5 | 1.17 | 15.56 | 88.8 |
| 2 | YT93159©® | YN73204 | CP721210 | 1 | 4.28 | 1.84 | 1.83 | 2.33 | 1.33 | 1.5 | 1.5 | 16.47 | 89.79 |
| 3 | YT89240©® | CP721210 | GT73167 | 1 | 3.49 | 1.27 | 1.67 | 2.17 | 1.33 | 2 | 1.5 | 15.38 | 88.18 |
| 4 | LC031137©® | HOCP93746 | ROC22 | 3 | 2.52 | 0.81 | 1.5 | 2.33 | 2 | 2.17 | 1.67 | 15.44 | 87.83 |
| 5 | YZ0680©® | ROC25 | CP723591 | 3 | 1.94 | 0.61 | 2 | 2.17 | 1.83 | 2.33 | 1.67 | 15.7 | 85.65 |
| 6 | ROC22©® | ROC5 | 69-463 | 3 | 0.83 | 0.76 | 2 | 2 | 2.5 | 2.5 | 2.33 | 15.55 | 87.48 |
| 7 | GT84332©® | HN5612 | NJ59782 | 3 | 0.76 | 0.53 | 1.83 | 3.33 | 1.83 | 2.67 | 2.17 | 15.59 | 86.52 |
| 8 | F172©® | F153 | F152 | 3 | 2.26 | 0.02 | 1.33 | 1.67 | 3.83 | 2 | 2.33 | 14.11 | 86.46 |
| 9 | DZ0383©® | YT85177 | ROC22 | 3 | 2.2 | 0.02 | 1.83 | 2.17 | 2 | 2.17 | 1.67 | 15.44 | 85.64 |
| 10 | YT00318©® | YN73204 | CP861633 | 3 | 1.94 | 0.23 | 2.17 | 2.17 | 2 | 2 | 1.83 | 15.02 | 88.08 |
| 11 | YZ6424©® | CO419 | Dongguawa2878 | 3 | 1.77 | 0.4 | 2.33 | 2.33 | 1.17 | 2.5 | 2 | 15.49 | 86.71 |
| 12 | YZ897©® | Xuan15 | YC84125 | 3 | 1.75 | 0.11 | 1.5 | 2.17 | 1.67 | 2.67 | 2.17 | 15.19 | 86.18 |
| 13 | GT96211©® | Pindar | Gt96167 | 3 | 1.31 | 0.36 | 2.33 | 2.5 | 1.83 | 2.33 | 1.83 | 15.36 | 87.17 |
| 14 | GT94119©® | GZ7565 | YC71374 | 3 | 0.89 | 0.47 | 1.83 | 2.5 | 1.83 | 2.83 | 2.17 | 15.17 | 86.51 |
| 15 | GT11©® | CP4950 | CO419 | 3 | 0.86 | 0.06 | 2 | 2 | 2.17 | 2.5 | 2.5 | 15.11 | 86.88 |
| 16 | YZ091601©® | CP941100 | CT89103 | 4 | 1.74 | 2.62 | 2.33 | 3.33 | 1.33 | 2.17 | 1.83 | 17.16 | 88.92 |
| 17 | ROC16©® | F171 | 74-575 | 4 | 1.34 | 1.36 | 2 | 1.67 | 2 | 2.5 | 2.33 | 15.87 | 87.39 |
| 18 | GT895©® | GT73167 | YC6240 | 4 | 1.12 | 1.14 | 2.17 | 2.17 | 2 | 2.5 | 2 | 15.73 | 87.92 |
| 19 | ROC23©® | F177 | 74-575 | 4 | 1.05 | 1.22 | 2 | 2 | 2 | 2.83 | 2 | 15.41 | 86.77 |
| 20 | LC05136©® | CP811254 | ROC22 | 4 | 0.91 | 1.27 | 2.17 | 2.17 | 1.67 | 2.67 | 2.33 | 15.95 | 88.45 |
| 21 | DZ9388©® | YC71347 | CP721210 | 4 | 0.65 | 2.21 | 2.5 | 3 | 2.17 | 2.33 | 2 | 16.59 | 87.61 |
| 22 | GZ7565©® | GZ64137 | NJ57416 | 4 | 0.38 | 1.77 | 1.83 | 3.17 | 3.17 | 2.17 | 2.83 | 16.22 | 88.38 |
| 23 | GN81711©® | CP67-412 | YC62-70 | 4 | 1.4 | 1.38 | 1.5 | 2.33 | 2.33 | 2.67 | 2 | 15.37 | 87.44 |
| 24 | GZ0270©® | GT69435 | CP841198 | 4 | 0.37 | 2.22 | 2.33 | 2.83 | 1.83 | 2.83 | 2.17 | 16.64 | 88.74 |
| 25 | MT69421©® | CP33310 | F134 | 4 | 0.49 | 1.24 | 2.17 | 2.83 | 1.83 | 2.67 | 2.33 | 15.87 | 88.67 |
| 26 | YT00236©® | YN73204 | CP721210 | 4 | 0.22 | 0.91 | 2.5 | 2 | 2.17 | 2.83 | 2.5 | 15.84 | 88.05 |
| 27 | YZ091028© | YR05178 | MT862121 | 3 | 3.42 | −0.13 | 1.83 | 2 | 1.67 | 1.67 | 1.67 | 14.98 | 87.03 |
| 28 | F170© | COL9 | PT54CP182 | 3 | 2.81 | −1.26 | 1.67 | 2.67 | 1.5 | 2.17 | 1.5 | 13.82 | 86.05 |
| 29 | GN912© | CP76380 | CP78304 | 3 | 1.77 | −4.06 | 1.83 | 2.67 | 1.67 | 2.33 | 1.83 | 12.97 | 80.62 |
| 30 | YZ082060© | YT93159 | Q121 | 3 | 1.35 | −0.48 | 1.67 | 2.33 | 2 | 2.5 | 2.17 | 14.79 | 90.78 |
| 31 | YZ011413© | YT85177 | ROC10 | 3 | 1.32 | −1.28 | 2 | 1.83 | 2 | 2.67 | 2 | 14.58 | 84.4 |
| 32 | GN01112® | G7565 | CP57614+ZZ82339 | 4 | −1.3 | 3.16 | 2.17 | 3.83 | 4 | 3 | 3.5 | 16.39 | 88.46 |
| 33 | LC03182® | CP721210 | ROC22 | 4 | −0.66 | 2.91 | 2.67 | 2.5 | 2.83 | 2.83 | 3 | 16.78 | 88.82 |
| 34 | ROC20® | 69-463 | 68-2599 | 4 | −0.19 | 2.7 | 2.67 | 2 | 2.67 | 3 | 2.5 | 16.57 | 89.8 |
| 35 | GN008® | CP57614 | YC84125 | 4 | −2.04 | 2.59 | 2.83 | 4.33 | 3 | 3.67 | 3.83 | 16.8 | 88.37 |

**Table 5.** *Cont.*

| Number | Varieties | Female Parent | Male Parent | Group | A_P_Value | T_P_Value | Height | Leaf Disease | Stem Diameter | Millable Stalks | General Vigor | Cane Sugar/% | Gravity Purity/% |
|---|---|---|---|---|---|---|---|---|---|---|---|---|---|
| 36 | GN89131® | G82660 | CP721210 | 4 | −2.06 | 2.46 | 3.17 | 4 | 3.83 | 3.33 | 3.67 | 16.1 | 86.84 |
| 37 | YZ02588® | CP721210 | YT843 | 4 | −1.23 | 2.32 | 2.67 | 3.83 | 2.5 | 3.17 | 3.17 | 16.84 | 88.86 |
| 38 | ROC8® | F146 | F160 | 4 | −1.97 | 2.3 | 3.17 | 3.5 | 3.33 | 3.5 | 3.67 | 16.63 | 88.71 |
| 39 | YZ022540® | ROC11 | CP723591 | 4 | −0.84 | 2.29 | 2.67 | 3 | 2.83 | 2.83 | 3 | 16.34 | 89.03 |
| 40 | YZ111779® | CP841198 | YZ94343 | 4 | −0.28 | 2.11 | 2 | 2.83 | 2.83 | 2.83 | 2.83 | 16.66 | 86.91 |
| 41 | FN30® | CP841198 | ROC10 | 4 | −0.14 | 2.03 | 2.67 | 2.5 | 2.83 | 2.67 | 2.33 | 16.09 | 88.71 |
| 42 | GT02901® | ROC23 | CP841198 | 4 | −0.91 | 1.98 | 2.67 | 2.67 | 2.67 | 3.17 | 3 | 16.17 | 87.33 |
| 43 | YZ05226® | ZZ74141 | CP721210 | 4 | −0.7 | 1.56 | 2.83 | 2.33 | 2.5 | 3.17 | 2.83 | 16.32 | 88.91 |
| 44 | FN38® | YT83257 | YT83271 | 4 | −0.11 | 1.38 | 2 | 2.67 | 2.5 | 2.83 | 2.83 | 15.65 | 87.74 |
| 45 | YZ0549® | YC9056 | ROC23 | 4 | −0.23 | 1.06 | 2.17 | 2.83 | 3 | 2.67 | 2.67 | 15.75 | 87.39 |
| 46 | FN39® | YT91976 | CP841198 | 4 | −0.31 | 1.04 | 2 | 3.33 | 2.17 | 3.17 | 2.67 | 15.57 | 87.83 |
| 47 | GT93102® | GT73167 | YC73512 | 5 | −2.33 | 1.78 | 3.17 | 4.5 | 3.67 | 3.67 | 3.83 | 16.06 | 85.75 |
| 48 | ROC26® | 71296 | ROC11 | 5 | −1.26 | 1.47 | 2.83 | 3 | 2.83 | 3.5 | 2.83 | 16.09 | 87.97 |
| 49 | LC07150® | YT85177 | ROC22 | 5 | −1.83 | 1.37 | 2.83 | 3.5 | 2.67 | 4 | 3.5 | 15.83 | 87.73 |
| 50 | GT60149® | Co290 | CP49-50 | 5 | −1.47 | 1.17 | 3 | 2.67 | 3.67 | 3 | 3.83 | 15.51 | 88.45 |
| 51 | GN79216® | NCo310 | CP44-101 | 5 | −1.24 | 1.09 | 2.83 | 2.67 | 3.33 | 3 | 3.5 | 15.59 | 87.06 |

"©" represents excellent agronomic traits, "®" represents excellent industrial traits, "©®" represents both excellent agronomic and industrial traits. "A_P_value" represents the agronomic projection value. "T_P_value" represents the industrial projection value.

## 4. Discussion

### 4.1. Projection Direction and Projection Values of Agronomic and Industrial Traits

Projection direction analysis.

The projection direction can be understood as the weight of a trait's influence on the whole evaluation. In the application of the PPC model, the determination of projection direction was a key point that directly determined the accuracy of the evaluation results [24]. In this experimental study, the general vigor (comprehensively evaluated and graded by other agronomic traits) grading data correlated significantly with the plant height, stem diameter, and millable stalks. When determining the projection direction, the PPC model fully considered the correlation relationship, and provided reasonable projection directions of the four traits, in which the direction of millable stalks was the highest, followed by the plant height and stem diameter, with the general vigor in the middle. From the difference comparison of leaf disease cluster groups (Figure 6), we found that there was no significant difference among cluster 1, cluster 3, and cluster 4, and there was no significant difference between cluster 2 and cluster 5. This indicated that the influence of leaf disease on variety differentiation was less than that of other traits; therefore, the projection pursuit model gave this trait the lowest projection direction (evaluation weight). From the above analysis, we considered that the determinations of the projection direction of agronomic traits were correct and in line with the data characteristics and evaluation system.

For the projection values of industrial traits, the gravity purity of sugarcane was the largest, followed by cane sugar, juice sugar, and brix, with fiber as the lowest (Figure 3). The projection direction of cane sugar and juice sugar were the same, and the brix projection direction was smaller. Pearson correlation analysis (Figure 2) showed that there was a significant positive correlation between sugarcane sucrose and juice sucrose, and the correlation coefficient reached 0.96. Therefore, when determining the projection direction, sugarcane sucrose content and cane juice sucrose content are equivalent. In sugarcane variety breeding, sugarcane brix can be used as a reference that reflects the sugarcane sugar content that cannot be ignored, which makes a significant difference in sugar content and recovery, and the projection direction of sugarcane brix was next to juice sucrose and sugarcane sucrose. As can be seen from Figure 6, there was no significant difference in sugarcane fiber content among the five sugarcane groups. Therefore, the influence of sugarcane fiber content on variety differentiation is poor, and the projection direction was the lowest. Except for no significant difference between cluster 1 and 4 groups of the gravity purity groups, there were significant differences among the other groups, which had a marked impact on variety differentiation; therefore, the projection direction was the highest. This also showed that the projection direction of sugarcane industrial characteristics was correct.

Projection value analysis.

After determining the correct projection direction, reliable projection values can be obtained. The projection value reflects the evaluation of the comprehensive performance of the object after the analysis of high-dimensional data [25]. Comprehensive evaluation of the research object according to the projection value has been widely used in hydrological research [26], but rarely in crops [27]. In the evaluation of sugarcane varieties, we performed preliminary exploration and research, forming the foundation for future studies [15]. In this study, the projection values of the industrial and agricultural characters of 103 sugarcane variety resources were widely distributed in the area of −5 to 5 (Figure 3), and the projection value of each sugarcane variety resource did not overlap. From Figure 4, we also found that the projection values of the industrial and agricultural characters of 103 sugarcane variety resources were well-distributed into four areas, and these four types of projection directions had different characteristics. This showed that the projection value can reflect the comprehensive performance of sugarcane variety resources well.

*4.2. Industrial and Agronomic Traits Separately Analyzed by PPC*

The PPC model has significant advantages in the analysis of high-dimensional data with discontinuous and non-normal distribution characteristics. From the frequency distribution analysis (Figure 1), we noted that the agronomic character grading data of the 103 sugarcane variety resources were discontinuous and non-normal distribution, whereas the industrial character traits represented continuous data, and, except for the juice sugar of sugarcane, the other traits showed normal distribution characteristics. Therefore, it was necessary to explore and analyze the industrial and agricultural traits separately to observe whether the determination of projection direction was in line with the expectations and breeding experience. The projection directions of both the industrial traits with a continuous normal distribution and the agronomic traits with discontinuous and non-normal distributions reached the expected results, and the evaluation result was accurate. In addition, in the evaluation and application of sugarcane variety resources, not all sugarcane variety resources or parents are considered to have better industrial and agricultural traits. Some varieties had excellent industrial characters and some varieties had excellent agronomic characters. In sugarcane breeding, we can improve a certain character through the selection of hybrid combinations to improve sugarcane varieties. Therefore, the separate study of industrial and agricultural traits was in line with the practice of modern sugarcane breeding, and could obtain an objective and comprehensive evaluation and analysis of variety resources.

*4.3. Analysis of the China Sugarcane cross Breeding from the Screened Variety Resources*

In 2020, the sugarcane planting area in Yunnan represented approximately 2.73 million hectares. ROC22 represented around 45.3 thousand hectares, YT93159 around 44.5 thousand hectares, LC05136 around 24.1 thousand hectares, YZ081609 around 12.67 thousand hectares, LC031137 around 7.27 thousand hectares, YT00236 around 9.46 thousand hectares, LC03182 around 3.7 thousand hectares, and YZ091601 around 133.3 hectares. These sugarcane varieties screened in Table 5 therefore accounted for above 70% of the total cultivated area in China.

Sugarcane yield and sucrose are the core contents of sugarcane variety improvement. Increasing yield and achieving a breakthrough in sucrose content are still the core tasks of sugarcane breeders. However, sugarcane disease has become a prominent problem of sugarcane improvement [28,29]. From Table 5, we found that variety YZ091601, which had excellent industrial and agronomic traits, but had been eliminated because of its higher comprehensive incidence rate than other varieties, with a leaf diseases average grade of 3.33 (an incidence rate of about 20%). Variety ROC22 (Table 5) was once a dominant sugarcane variety in China. However, after long-term cultivation, its ratoon smut was serious, and its production area is now less than 30%. As a representative of the new generation of improved sugarcane varieties, YZ081609 [30] has excellent sugar content and yield, and is deeply appreciated by sugar enterprises and sugarcane farmers. It has become representative of the main sugarcane varieties promoted and planted in the Yunnan sugarcane area at present. Variety YZ011413 [31] was not popular because of its high natural incidence of rust; however, it also has good performance for agronomic traits (Table 5). Many varieties in Table 5 screened by the PPC were consistent with the actual performance of varieties in production.

From Table 5, we also found that the genetic relationship of these main production varieties is similar, and the genetic basis is narrow [32]. The establishment of a sugarcane hybrid parent system in China has occurred through the combination of introduction and independent innovation. Foreign germplasm resources have played an important role in the breeding of new sugarcane varieties in China. According to incomplete statistics, 163 varieties were bred from 21 parents, such as introduced germplasm CP72120, CP49-50, F134, and Co419, accounting for 87.63% of the counted varieties, since the 1950s in China [33]. CP, F, and Co series germplasm have become the real core parents in China [34]. The genetic backgrounds of the main sugarcane varieties are very similar, with many varieties sharing

common parents or common kinship relationships. The commercial varieties with excellent performance in China sugarcane areas generally have kinship relationships, such as CP, F, Co, and YC (Table 5).

## 5. Conclusions

The PPC model evaluated and analyzed the variety resources well, as well as the discontinuous and non-normally distributed agronomic traits. The judgment on the projection direction of industrial and agronomic traits met the breeding experience and expectation, and accounted for the diversity of data. The projection values were consistent with the comprehensive performance of commercial sugarcane varieties, and could be well-distinguished. The application of PPC in the comprehensive evaluation of sugarcane variety resources is feasible and can be explored as a new evaluation method. At the same time, according to the evaluation and analysis of 103 commercial varieties based on the PPC model, 51 excellent industrial and agronomic variety resources were screened, which could be used as parental references for sugarcane hybrid breeding.

**Author Contributions:** Writing—original draft, Y.Z. (Yong Zhao); Data curation, F.Z. and J.Z.; Experiments, P.Z. and J.D.; Funding acquisition, Y.Z. (Yuebin Zhang) and C.W.; Writing—review and editing, J.L. All authors have read and agreed to the published version of the manuscript.

**Funding:** This work was supported by the National Natural Science Foundation of China (grant numbers 32060505, 31660418); the National Sugar Industry Technology System (grant number CARS-170101); the Yunnan Science and Technology Planning Project, China (grant number 202101AT070273); the Yunnan Overseas High-level Talent Introduction Program, China (grant number GDWG-2018-015); the Yunnan Science and Technology Support the Development of Green Industry, China (grant number 202004AC100001-A02); the Sugarcane Germplasm Innovation and New Variety Breeding Team of Yunnan Academy of Agricultural Sciences, China (grant number 2019HC013); and the Yunnan Key Research and Development Projects, China (grant number 2019IB008). The authors thank the Guangxi Key Laboratory of Sugarcane Genetic Improvement Program (21-238-16) for supporting Yong Zhao.

**Institutional Review Board Statement:** Not applicable.

**Informed Consent Statement:** Not applicable.

**Data Availability Statement:** The authors confirm that the data supporting the findings of this study are available within the article.

**Acknowledgments:** We thank Zhiyuan Wang and Panlei Wang for assistance of the data analysis and the use of the R software.

**Conflicts of Interest:** The authors declare no conflict of interest.

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
