# Peer review of "New Method for Sugarcane (Saccharum spp.) Variety Resources Evaluation by Projection Pursuit Clustering Model"

_agronomy, doi:10.3390/agronomy12061250_

Round 1

Reviewer 1 Report

Dear Authors 

The following are my concerns regarding the manuscript titled " Projection Pursuit Clustering Model Applied to Evaluate Sugarcane (Saccharum spp.) Variety Resources Bred in the Last 70  Years in China"

  1. The details R package details were not given in material and methods
  2. What is the purpose of ranking for quantitative traits like height, diameter of millable stalks? Rank scoring is ok for disease susceptibility/resistance.
  3. What is the basis of ranking whether score 1 is significantly different from score 2?
  4. The QQ plot of scores does not represent quantitative traits, as it is not continuous and must be removed from the manuscript. Frequency distribution can be used instead.
  5. In case of PPC analysis for agronomic traits whether the scores were used or the metric data? Not clearly mentioned in the manuscript.
  6. The correlation studies must be done with quantitative data not with the scores. The scores have a range.
  7. In the projection value distribution chart, the traits are not mentioned.
  8. Agronomic and technological projection values are not defined anywhere in the manuscript.
  9. Does industrial and technological projections value mean the same?
  10. In the discussion part "Therefore, when determining the projection direction, sugarcane sucrose content and cane juice sucrose content are equivalent. In sugarcane varietal breeding, sugarcane brix is only used as a reference that reflects the sugar content, and is not a necessary an index, thus the projection direction is lower  than that of the sugar content." objectionable because sugarcane is grown for sugar, estimated  by Brix value. Brix is an important trait in breeding which cannot be ignored. Brix of 1 and 2 may not show significant difference statistically in the sample but it makes a significant difference in sugar content and recovery.
  11. For comparison any other multivariate analysis like MGIDI must be performed to assess the effectiveness of PPC for selection.

Reviewer 2 Report

The manuscript discusses interesting ideas and data but can not be evaluated accurately due to limitations. 

The title needs revision. Please change your title according to your objective and conclusion. I suggest that you provide a clear objective for your study. 

A major drawback is that the authors provide very limited information on the field trial. Only one location and two replications from what I understood. Authors only provide 7 lines to discuss this essential information.

Please provide comprehensive and detailed information on the experimental design, the number of treatments, the number of replications used, the size of the plots, the no. of rows per plot. Only one location and two replications may not be enough to provide accurate data for the statistical modeling and for making valid conclusions. 

Round 2

Reviewer 2 Report

Authors have improved the manuscript and have responded to my questions. 
